# Understanding the impact of moxifloxacin on immune function: Findings from cytokine analyses and immunological assays in mice

Mehmood Ahmad[1,2]*, Naeem Rasool[2], Rana Muhammad Zahid Mushtaq[3], Sammina Mahmood[4], Abdur Rauf Khalid[5], Waqas Ahmad[6,7], Bilal Mahmood Beg[2], Mostafa A. Abdel Maksoud[8], Abdul Aziz Alamri[9], Adeel Sattar [2]*

**1** Department of Pharmacology and Toxicology, Faculty of Veterinary and Animal Sciences, The Islamia University of Bahawalpur, Bahawalpur, Pakistan, **2** Department of Pharmacology and Toxicology, University of Veterinary and Animal Sciences, Lahore, Pakistan, **3** Division of Infection Medicine, College of Medicine, The University of Edinburgh, Edinburgh, United Kingdom, **4** Department of Botany, Division of Science and Technology, Bank Road Campus, University of Education, Lahore, Pakistan, **5** Department of Livestock and Poultry Production, Faculty of Veterinary Sciences, Bahauddin Zakariya University, Multan, Pakistan, **6** Department of Pathology, University of Veterinary and Animal Sciences, Lahore, Pakistan, **7** Livestock and Diary Development Department, Government of Punjab, Lahore, Pakistan, **8** Botany and Microbiology Department, College of Science, King Saud University, Riyadh, Saudi Arabia, **9** Biochemistry Department, College of Science, King Saud University, Riyadh, Saudi Arabia

* mehmood.ahmad@iub.edu.pk (MA); adeel.sattar@uvas.edu.pk (AS)

## Abstract

(1) Background: Moxifloxacin (MXF) is a fluorinated quinolone antibiotic used most commonly due to its broad spectrum of activity. However, the immunomodulatory effects of MXF remain underexplored. This study aims to investigate the patterns of immunomodulatory effects. (2) Methods: Swiss female albino mice were treated with different concentrations of MXF, and the immunological studies were performed using a cytokine assay, carbon clearance test, indirect hemagglutination test, and a mice lethality assay. (3) Results: Results revealed that MXF mitigated cyclophosphamide-induced leukopenia, with TLC reduced less at 7.5 mg/kg (5.37 ± 0.6) compared to 3.75 mg/kg (2.95 ± 0.485) and 15 mg/kg (2.14 ± 0.104). Results of the phagocytic index showed significantly lower clearance rates in the 7.5 mg/kg (0.01600 ± 0.00175) and 15 mg/kg (0.01331 ± 0.00310) groups compared to the control. Ex-vivo mice macrophages cytokines analysis showed elevated TGF-β1 levels in the MXF 32 µg/ml group (30.826 ± 0.817 pg/ml) and IL-10 levels in the MXF 16 µg/ml group (50.427 ± 0.786 pg/ml). The MXF 64 µg/ml treated group showed significantly lower IL-6 (8.714 ± 0.647 pg/ml) and TNF-α (28.81 ± 3.24 pg/ml) levels compared to controls. Findings of indirect hemagglutination assay showed higher antibody titers in the 3.75 mg/kg group (7.200 ± 0.374) compared to higher doses, whereas dose-dependent mortality was observed, with the highest mortality at 15 mg/kg (80%) in the mice lethality assay. (4) Conclusions: MXF exhibited notable immunomodulatory effects by mitigating cyclophosphamide-induced immunosuppression, modulating

**Data availability statement:** All relevant data are within the manuscript and its Supporting Information files.

**Funding:** The author(s) Mostafa A. Abdel Maksoud and Abdul Aziz Alamri received funding for this work from project number (RSPD2025R552), King Saud University, Riyadh, Saudi Arabia.

**Competing interests:** The authors have declared that no competing interests exist.

cytokine levels, phagocytic activity, and antibody production. The findings suggested that MXF could be considered for managing immunosuppression.

## 1. Introduction

Fluoroquinolones (FQs) are synthetic antibiotics active against many genera of bacteria, both the gram-positive and the gram-negative [1]. Of all generations of quinolones, there is increased evidence that fourth-generation FQs have better activity against anaerobes and pneumococci than the other generations [2]. Moxifloxacin (MXF) has been reported to have respiratory tract infection activity, particularly against multi-drug-resistant pneumococcal strains and multiple anaerobes, including Mycobacterium tuberculosis [3]. Antibiotics are often employed in treating the condition in adults for the respiratory tract, skin, and urinary tract infections [4]. MXF is also used in the second-line treatment of TB if the patient does not show any response to other first-line drugs [5]. Current literature has also documented that all FQs exhibit immunomodulatory effects in the host apart from the bacteriostatic effect [6].

In addition to bacteriostatic, the latest published research has shed light on the immunomodulatory effects of many FQs in the host [6]. The chemical structure shows that MXF has cyclopropyl moiety positioned at position 1, indicating that the drug has immunomodulatory activity. It is well documented that MXF increases the synthesis of granulocyte colony-stimulating factor (G-CSF), thus stimulating hematopoiesis and increasing the immune response [7]. These immunomodulatory effects are regarded as provoking the synthesis of cytokines and growth factors, for instance, IL-2, IL-3, interferon-γ and-I, and TGF-β [8]. MXF works as an immuno-modulatory agent, it has been reported to reduce IL-1β and TNF-α levels in immune cells which were stimulated with and lipopolysaccharides. Furthermore, MXF also play protective role in the murine model of sepsis. Some published investigations show that some cytokines produced by human and murine WBC can be affected by MXF [9]. As for the impact of FQs on cytokines, it demonstrates no consistency and depends on the concentration of the drug. For example, ciprofloxacin and MXF have been demonstrated to alter the synthesis of cytokines and growth of the hematopoietic cells; concentrations above the pharmacokinetic range were stated to be inhibitory to the murine and human hematopoietic cell line models [10].

TGF-beta1, IL-10, IL-6, and TNF-alpha cytokine profiles are considered regulatory cytokines that play a crucial role in immune response [11]. Various isoforms of TGF-beta help control lymphocytes, dendritic cells, natural killer cells, and macrophages, regulating them to avoid autoimmune diseases [12]. IL-10 is an anti-inflammatory cytokine capable of suppressing the production of inflammatory cytokines like TNF-alpha, IL-1, and IL-6, thus playing immune regulatory roles, and protecting tissues from injury [13]. Different cells secrete IL-6, and its primary function is to elicit an immune response through the JAK/STAT pathway. Excessive secretion of IL-6 is associated with neoplasia and chronic inflammatory diseases [14]. TNF-alpha is involved in systemic inflammation and the acute phase reaction, control of immune cells, fever, apoptotic cell death, and sepsis; autoimmune diseases, insulin

resistance, and cancer are some conditions in which the levels of TNF-alpha are increased [15]. Literature regarding the mechanism of MXF and how it alters these cytokines is crucial in understanding immune regulation.

The details of the immunomodulatory effects of MXF and its application in noninfectious inflammatory diseases have not been investigated enough. Further, the effects of MXF on immune system regulation, in particular, the changes that remain in the long term, were not investigated. To the best of our knowledge, this study is the first to attempt the investigation of the immunomodulatory activity of MXF employing Swiss albino mice as animal models. The study aims to use *in-vitro* and *in-vivo* techniques to establish the immunological effects of MXF.

## 2. Materials and methods

### 2.1. Reagents

MXF HCl (batch no. MOXI/1803005, analytical reference no. FP/2018–19/039) was obtained from Aarti Drugs Limited, India.

### 2.2. Experimental animal and protocol

Swiss female albino mice (BALB/c), 25-30g and 6–8 weeks old, were procured from the Department of Theriogenology, University of Veterinary and Animal Sciences (UVAS) Lahore, Pakistan. Mice were acclimated for 5–7 days before the experimental trials, housed under a 12-hour light/dark cycle with controlled temperature and humidity, and provided with a standard pellet diet and clean water ad libitum [16]. The study protocol was approved by the Institutional Review Board of University of Veterinary and Animal Sciences Lahore, Pakistan (No. DR/921 dated. 04-09-2023). Ethical guidelines were followed as per the institutional ethical committee and in accordance with ARRIVE guidelines.

Since the study involves the use of mice, the humane endpoints were kept under consideration throughout the study [17,18]. All the mice that were used in the experimental protocols were handed back to the Department of Theriogenology, UVAS, Lahore. The mice were observed over a period of 3 days to a week for rapid weight loss, labored breathing, paralysis or prostration, and in cases where the moribund animal was observed, it was anesthetized [19] and then euthanized with cervical dislocation by a registered veterinarian [20]. Cervical dislocation was chosen as a preferred method as it is permissible as per the guidelines issued by the University of Veterinary and Animal Sciences protocol as well as guidelines laid out by AVMA for the euthanasia of animals [21]. Moreover, while designing the study the minimal statistical limit was kept into consideration and the least number of mice were taken for the assays where the death-endpoints were likely, such as mice lethality assay.

### 2.3. Cyclophosphamide induced neutropenic assay

Twenty adult mice were divided into control positive and negative groups along with three MXF-treated groups (3.75 mg/kg, 7.5 mg/kg, 15 mg/kg). Control positive group only received cyclophosphamide dose, whereas the negative group did not receive any treatment. Each treated group had five mice, receiving MXF intraperitoneally thrice daily for ten days. On the 11th day, blood samples were collected to measure total leukocyte count (TLC) and differential leukocyte count (DLC). A neutropenic dose of cyclophosphamide (200 mg/kg, obtained from Clinix Pharmacy, Lahore) was then administered subcutaneously [22,23]. Blood (including serum) samples were collected again on the 13th day, and TLC and DLC were measured and compared. This test was employed to evaluate the effect of MXF on neutropenic mice.

### 2.4. Isolating peritoneal macrophages and determination of cytokines

Murine peritoneal macrophages were isolated by injecting 5 ml of ice-cold phosphate-buffered saline (PBS, Sigma-Aldrich) into the abdomen of euthanized mice and collecting the fluid [24]. The cells were resuspended in RPMI-1640 media with 10% FBS, incubated in 5% $CO_2$ for 3 hours, and removed non-adherent cells [25]. Adherent cells were treated with

various concentrations of MXF (4 µg/ml to 64 µg/ml) or distilled water (control group) for 24 hours. Supernatants were collected for various cytokines, including TGF-1β, IL-10, IL-6, and TNF-α quantification by ELISA.

TGF-1β, IL-10, IL-6, and TNF-α in the serum were measured using an ELISA kit (mouse TGF-β1, Cloud-Clone Corp.) following the manufacturer's instructions.

The cytokines analysis was performed to evaluate the anti-inflammatory effects of the MXF on the murine peritoneal macrophages [26].

## 2.5. Carbon clearance test

Carbon Clearance test was carried out to evaluate the phagocytic activity of the macrophages [27]. Twenty adult female albino mice were divided into four groups. The control group orally received carboxymethylcellulose (CMC, 0.5% solution, 10 ml/kg) for seven days. After 24 hours, 0.1 ml of Indian ink (batch no. 5306, code no. I-04818, Oxford Lab Chem, Mumbai, India) dispersion was injected into the tail vein of all mice [28]. Blood samples were taken at 0- and 15-minutes post-injection, mixed with 0.1% sodium carbonate solution, and absorbance was measured at 660 nm. The rate of carbon clearance (K) and phagocytic index (α) were calculated as previously described [29–32].

## 2.6. Hemagglutination assay

Mice were treated with MXF three times daily for 28 days. Sheep red blood cells (SRBCs, 1%) were injected to immunize the mice on the 14th and 21st days. Blood samples were collected on the 28th day, and hemagglutination (HA) titers were determined by measuring the lowest volume of serum that caused agglutination. The test provides the insights into detecting and quantifying the antibodies by the clumping of the SRBCs.

## 2.7. Mice lethality test

Mice lethality test was used to evaluate the effect of the MXF on the survival rate of mice following a challenging dose of *P. multocida*. A high survival rate of the mice would exhibit the immunostimulatory effect whereas the low survival would point out to the immunosuppressive effect due to the decreased IgG and IgM production [33]. Mice were treated with MXF at 3.75, 7.5, and 15 mg/kg intraperitoneally for 21 days. All mice were vaccinated with the hemorrhagic septicemia (HS) vaccine (gifted by the Veterinary Research Institute, Lahore, Pakistan) on the 7th and 17th days. On the 21st day, a challenging dose of *P. multocida* culture ($25 \times LD50$, $10^7$ cells/ml) was administered (except in control negative group). Mice were observed for three days, and the mortality ratio was calculated.

## 2.8. Statistical analysis

The data regarding continuous variables was checked for normality using the Kolmogorov-Smirnov test. Value of the IHA titers were subjected to $Log_2$ transformation. One-way ANOVA, followed by Tukey's post hoc test, was used to compare the mean values of diverse groups regarding various parameters. Results were presented as mean ± standard error of the mean (SEM). The data regarding the mice lethality test was presented as descriptive statistics and subjected to a Chi-square test with Montecarlo simulation in IBM SPSS version 25. Graphs were generated using GraphPad Prism version 9.5.

## 3. Results

### 3.1. In-vitro cytokines estimation

The mean TGF-β levels (pg/ml) varied significantly across treatment groups. As presented in Fig 1, the control group had a mean value of 24.129 ± 0.989 pg/ml. Notably, the highest mean TGF-β level was observed in the MXF 32µg/ml group at 30.826 ± 0.817 pg/ml. The MXF 16µg/ml group also exhibited elevated levels of TGF-β with a mean of 28.04 ± 1.05 pg/ml.

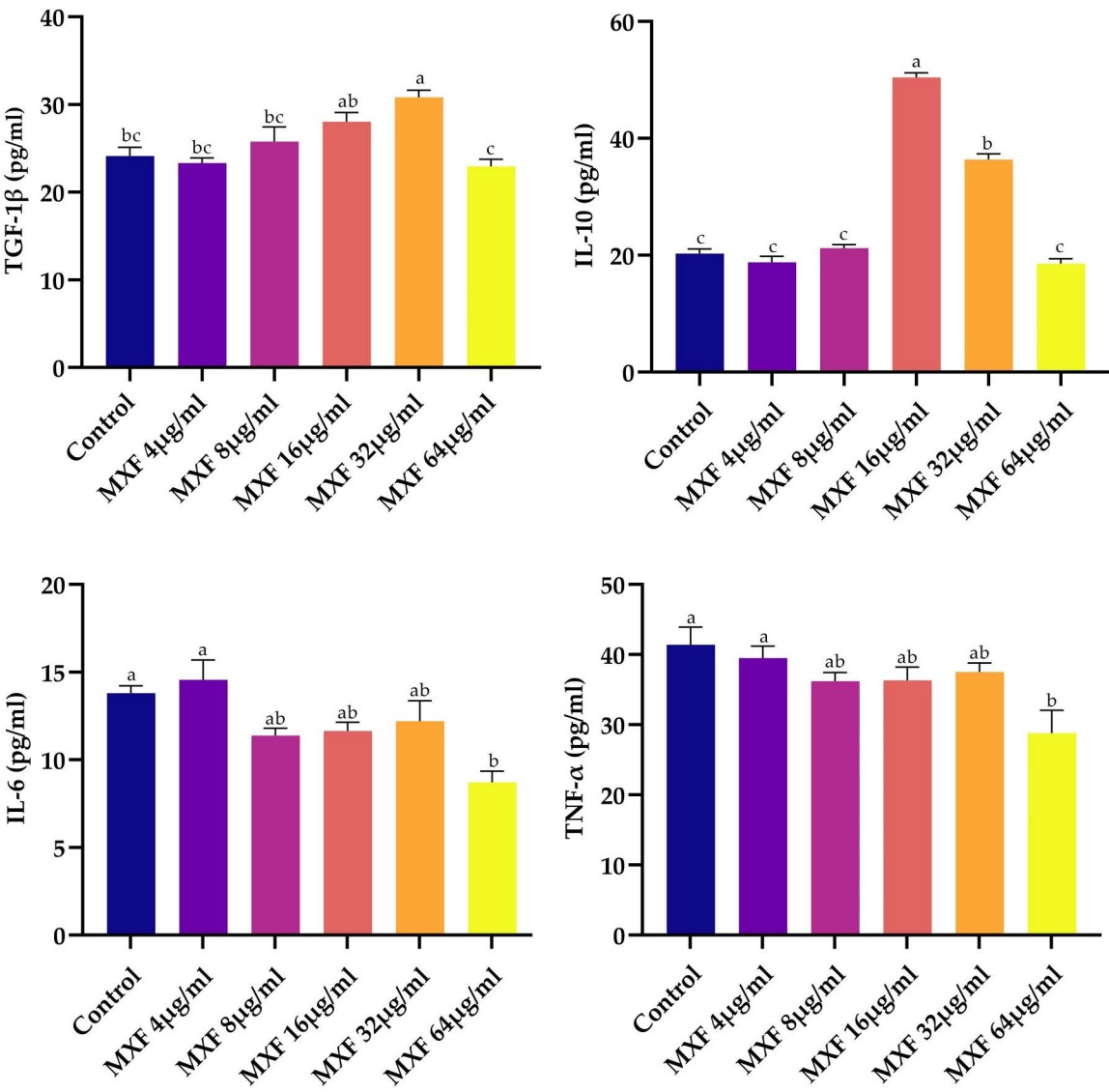

**Fig 1. Results of immunomodulatory effects of different concentrations of MXF on cytokine levels in murine ex-vivo peritoneal macrophages.**
The cytokines analyzed include TGF-β, IL-10, IL-6, and TNF-α. Bars presented as mean±SEM compared through one-way ANOVA followed by Tukey's post hoc test, with different superscripts indicate statistically significant differences (p<0.05) among groups.

Results indicated that the TGF-β levels in the MXF 32µg/ml group were significantly higher than those in the control, MXF 4µg/ml (23.319±0.606pg/ml), and MXF 64µg/ml (22.965±0.812pg/ml) groups. The MXF 16µg/ml treated group exhibited significantly higher mean IL-10 levels (50.427±0.786pg/ml), followed by the MXF 32µg/ml group (36.417±0.940pg/ml). Moreover, the IL-10 levels in the However, the mean IL-10 levels in the control group (20.325±0.753pg/ml), MXF 4µg/ml (18.83±1.01pg/ml), and MXF 64µg/ml (18.585±0.814pg/ml) groups did not vary significantly. For IL-6, the control group had a mean level of 13.805±0.412pg/ml, with the highest mean level observed in the MXF 4µg/ml (14.57±1.13pg/ml). Conversely, the MXF 64µg/ml (8.714±0.647pg/ml) group had the lowest IL-6 levels. Statistical analysis confirmed that the IL-6 levels in the MXF 64µg/ml group were significantly lower than in the control group. Additionally, TNF-α levels showed notable differences among the treatment groups. The control group had a mean TNF-α level of 41.40±2.56pg/ml.

The MXF 4μg/ml group showed a comparable mean TNF-α level of 39.51±1.70pg/ml. However, the MXF 64μg/ml group exhibited a significant reduction in TNF-α levels, with a mean of 28.81±3.24pg/ml. Statistical analysis revealed that the TNF-α levels in the MXF 64μg/ml group were significantly lower than those in the control group. Other MXF concentrations (16μg/ml, 32μg/ml, and 8μg/ml) showed intermediate TNF-α levels, which were not significantly different from the control group.

### 3.2 . Cyclophosphamide-induced neutropenic assay

The administration of cyclophosphamide significantly altered the total leukocyte count (TLC) across all treatment groups (Table 1). In the negative control group, the TLC decreased from 10.1±1.96 to 2.41±0.203 after treatment. MXF administration at 3.75mg/kg reduced TLC from 13.4±2 to 2.95±0.485. At 7.5mg/kg, the TLC decreased from 8.33±0.954 to 5.37±0.6. The highest dose of MXF (15mg/kg) showed a TLC reduction from 6.87±0.637 to 2.14±0.104. These findings indicate that MXF administration at various doses mitigated the cyclophosphamide-induced leukopenia, with the 7.5mg/kg dose showing a lesser reduction in TLC than other doses. The negative control group's lymphocyte count (103/μl) decreased significantly from 6.89±1.45 to 0.849±0.0965 after treatment. In the group treated with 3.75mg/kg of MXF, lymphocyte levels dropped from 8.24±1.33 to 1.53±0.234. For the 7.5mg/kg dose group, the count decreased from 4.48±0.497 to 2.48±0.331, while in the 15mg/kg dose group, the lymphocyte count was reduced from 3.95±0.303 to 1.19±0.0532. Monocyte counts also showed a decrease across all groups, with the negative control group declining from 0.62±0.114 to 0.132±0.0109, the 3.75mg/kg group from 0.681±0.0926 to 0.206±0.044, the 7.5mg/kg group from 0.471±0.0544 to 0.302±0.0341, and the 15mg/kg group from 0.386±0.0382 to 0.135±0.00912. Neutrophil counts (10³/μl) in the negative control group decreased from 2.55±0.405 to 1.42±0.104, in the 3.75mg/kg treated group from 4.31±0.58 to 1.2±0.221, in the 7.5mg/kg group from 3.33±0.433 to 2.56±0.285, and in the 15mg/kg group from 2.54±0.279 to 0.809±0.0744. Eosinophil counts (10³/μl) exhibited minor changes, with the negative control group showing a decrease from 0.0195±0.0195 to 0.0059±0.00391, the 3.75mg/kg group from 0.123±0.0308 to 0.00588±0.00588, the 7.5mg/kg group from 0.0515±0.0131 to 0.0315±0.0142, and the 15mg/kg group from 0.0387±0.0168 to 0.00856±0.00417.

### 3.3. Carbon clearance and phagocytic index

The control group had a mean clearance rate of 0.022386±0.000985 (Fig 2). Mice treated with 3.75mg/kg of MXF had a similar mean clearance rate of 0.02379±0.00207, showing no significant difference from the control group. However, the

**Table 1. Comparison of hematological parameters (TLC, Lymphocyte, Monocyte, Neutrophils, and Eosinophils) measured before and after cyclophosphamide treatment with different doses of MXF in comparison to the negative control. Data was compared through one-way ANOVA followed by Tukey's post hoc test.**

| Variable | Negative Control | | MXF 3.75 mg/kg | | MXF 7.5 mg/kg | | MXF 15 mg/kg | |
|---|---|---|---|---|---|---|---|---|
| | After | Before | After | Before | After | Before | After | Before |
| TLC (10³/mm3) | 2.41±0.203[c] | 10.1±1.96[ab] | 2.95±0.485[c] | 13.4±2[a] | 5.37±0.6[bc] | 8.33±0.954[ab] | 2.14±0.104[c] | 6.87±0.637[bc] |
| Lymphocyte (10³/ul) | 0.849±0.0965[d] | 6.89±1.45[ab] | 1.53±0.234 cd | 8.24±1.33[a] | 2.48±0.331 cd | 4.48±0.497[bc] | 1.19±0.0532 cd | 3.95±0.303[bd] |
| Monocyte (10³/ul) | 0.132±0.0109[d] | 0.62±0.114[ab] | 0.206±0.044 cd | 0.681±0.0926[a] | 0.302±0.0341 cd | 0.471±0.0544[ac] | 0.135±0.00912[d] | 0.386±0.0382[bcd] |
| Neutrophils (10³/ul) | 1.42±0.104 cd | 2.55±0.405[bc] | 1.2±0.221 cd | 4.31±0.58[a] | 2.56±0.285[bc] | 3.33±0.433[ab] | 0.809±0.0744[d] | 2.54±0.279[bc] |
| Eosinophils %(10³/ul) | 0.0059±0.00391[b] | 0.0195±0.0195[b] | 0.00588±0.00588[b] | 0.123±0.0308[a] | 0.0315±0.0142[b] | 0.0515±0.0131[ab] | 0.00856±0.00417[b] | 0.0387±0.0168[b] |

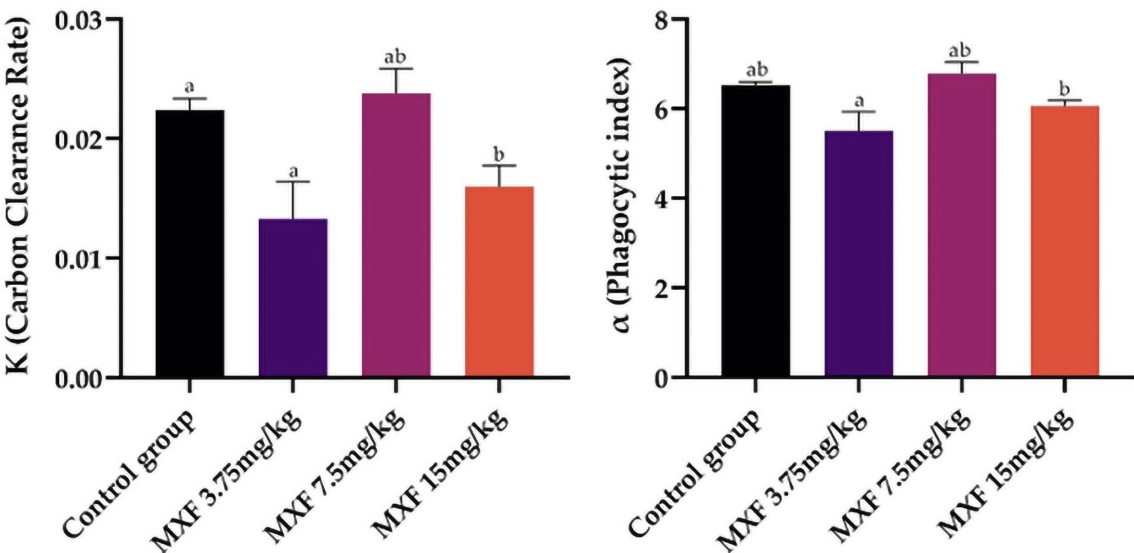

**Fig 2. Graphical illustration of Carbon clearance rate (K) and phagocytic index ( α) in Swiss albino mice treated with different doses of moxi-floxacin (MXF) compared to the control group.** Data are presented as the histograms (mean ± SEM) compared through one-way ANOVA followed by Tukey's post hoc test, having different superscripts indicate significant differences among the groups (p < 0.05).

7.5 mg/kg MXF group had a significantly lower mean rate of 0.01600 ± 0.00175 than the 3.75 mg/kg group. The 15 mg/kg MXF-treated group exhibited the lowest mean clearance rate of 0.01331 ± 0.00310, significantly different from the control and the 3.75 mg/kg groups. The phagocytic index (α) also varied significantly among the treatment groups. The control group exhibited a mean phagocytic index of 6.5251 ± 0.0738. Mice treated with 3.75 mg/kg of MXF showed a slightly higher mean phagocytic index of 6.782 ± 0.264, with no significant difference compared to the control group. In contrast, the group treated with 7.5 mg/kg of MXF had a lower mean phagocytic index of 6.064 ± 0.126, significantly different from the 3.75 mg/kg group but not the control group. The lowest mean phagocytic index was observed in the 15 mg/kg MXF group, measuring 5.501 ± 0.439, significantly lower than the 3.75 mg/kg.

### 3.4. In-vivo cytokines analysis

Analysis of TGF-β levels showed that the MXF 15 mg/kg and Positive Control groups had the highest mean values, 403.1 pg/ml and 383.5 pg/ml, respectively, with no significant difference (Fig 3). Both were significantly higher than the MXF 3.75 mg/kg group, with the lowest mean of 232.0 pg/ml. The MXF 7.5 mg/kg group, with a mean of 358.1 pg/ml, differed significantly from the Negative Control (253.3 pg/ml) but not from the MXF 15 mg/kg and Positive Control groups. For IL-10, the Positive Control group had the highest mean level at 48.205 pg/ml, significantly exceeding the MXF 15 mg/kg group's mean of 17.57 pg/ml. The MXF 3.75 mg/kg group, with a mean of 43.02 pg/ml, was similar to the Positive Control group but significantly higher than the MXF 7.5 mg/kg and Negative Control groups. The MXF 7.5 mg/kg group (30.78 pg/ml) was significantly higher than the Negative Control (20.345 pg/ml) but lower than the Positive Control and MXF 3.75 mg/kg groups. IL-6 levels were highest in the Negative Control group at 15.628 pg/ml, significantly higher than the MXF 15 mg/kg group at 8.57 pg/ml. The MXF 3.75 mg/kg (13.783 pg/ml) and Positive Control (13.572 pg/ml) groups showed no significant difference between each other but were significantly higher than the MXF 7.5 mg/kg group (11.21 pg/ml). Regarding TNF-α, the Negative Control group had the highest mean level at 40.31 pg/ml, significantly higher than the MXF 15 mg/kg group's mean of 20.99 pg/ml. The Positive Control (36.990 pg/ml) and MXF 3.75 mg/kg (35.49 pg/ml) groups showed no

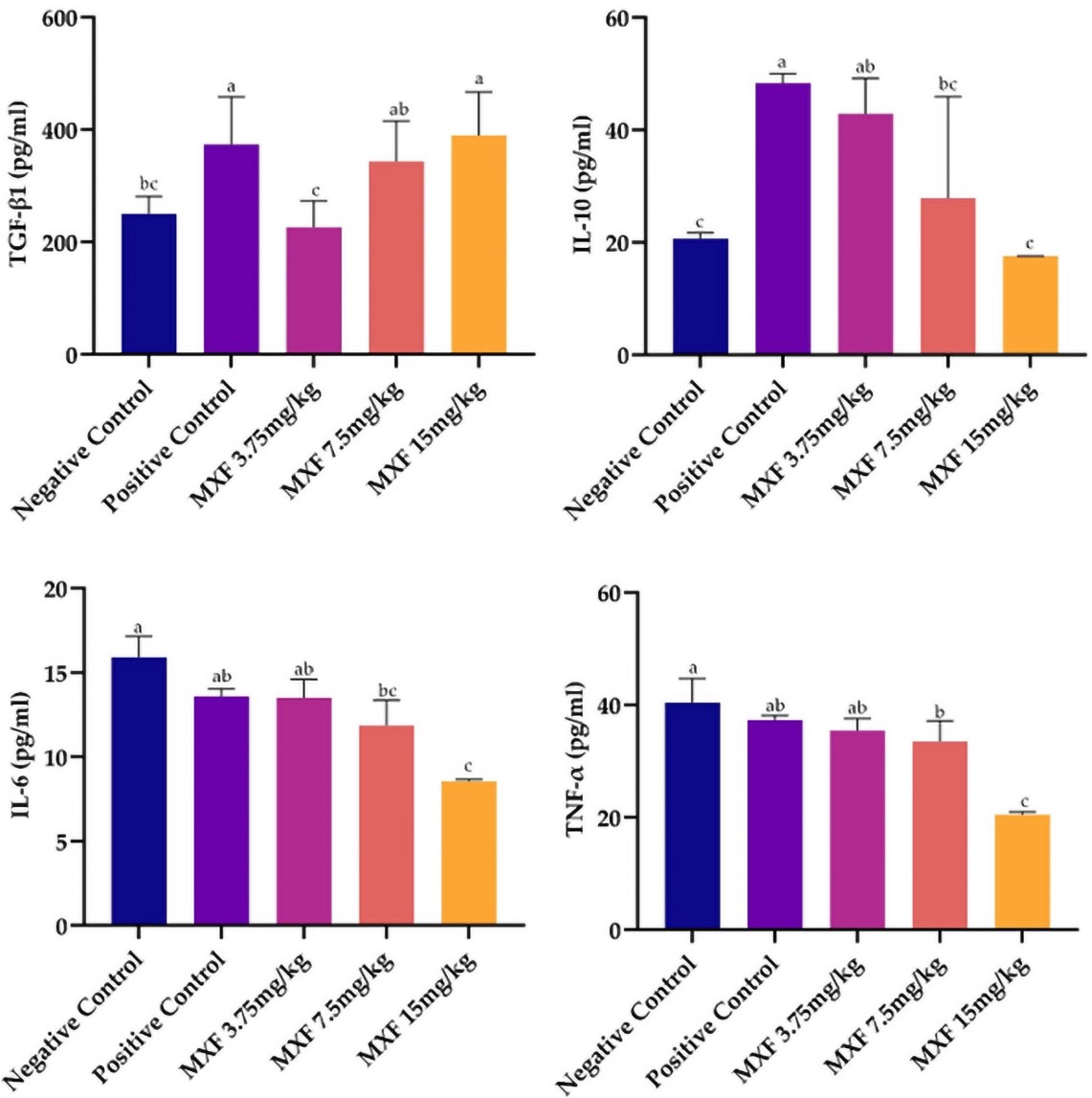

**Fig 3. Results of immunomodulatory effects of different concentrations of moxifloxacin MXF on cytokine levels in Swiss albino mice.** The cytokines analyzed include TGF-β, IL-10, IL-6, and TNF-α. Bars presented as mean±SEM compared through one-way ANOVA followed by Tukey's post hoc test, with different superscripts indicate statistically significant differences (p<0.05) among groups.

significant difference but were significantly higher than the MXF 7.5mg/kg group (33.84pg/ml). The MXF 15mg/kg group had significantly lower TNF-α levels than all other groups, indicating a substantial reduction at this higher dose of MXF.

### 3.5. Indirect hemagglutination assay

The Fig 4 presents that the positive control group had the lowest mean titer of 2.800±0.374, indicating the lowest antibody response. In contrast, the negative control group had a mean titer of 6.800±0.374, which was not significantly different from the 3.5mg/kg MXF group with a mean titer of 7.200±0.374. The 7.5mg/kg MXF group had a mean titer of 5.400±0.510, while the 15mg/kg MXF group had the lowest mean titer at 5.200±0.374. These two MXF groups (7.5mg/kg and 15mg/kg) showed significantly lower titers than the negative control and the 3.5mg/kg MXF groups.

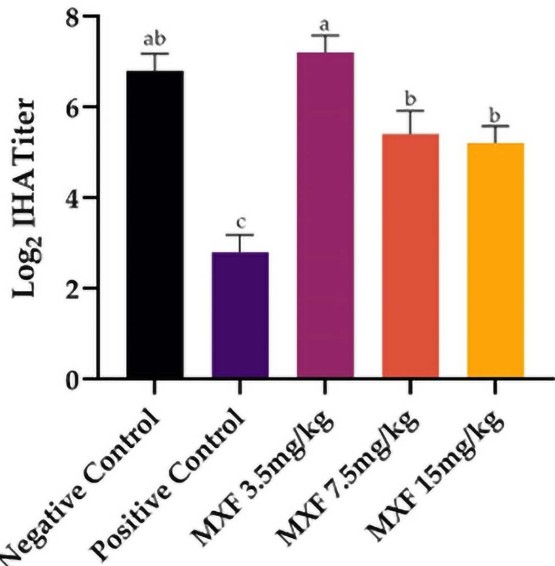

**Fig 4. Indirect Hemagglutination assay (IHA) titers (Log2) in Swiss albino mice showing significant differences across treatment groups with MXF 7.5 mg/kg dose revealing the highest titer and the positive control having the lowest (p < 0.05).** Data was compared through one-way ANOVA followed by Tukey's post hoc test.

## 3.6. Mice lethality assay

The mice lethality test results revealed a significant difference in mortality among the groups (p = 0.0173) (Table 2). In the negative control group, 0% mortality was observed. Conversely, the positive control group experienced a 100% mortality rate. The group receiving 3.75 mg/kg of MXF had a 40% mortality rate, while the group treated with 7.5 mg/kg of MXF exhibited a 60% mortality rate. The highest dose of 15 mg/kg of MXF resulted in the highest mortality rate among the treatment groups, with 80% of the mice deceased.

## 4. Discussion

This study aimed to evaluate the immunomodulatory profile of MXF in the Swiss albino mice model. MXF administration at 3.75 mg/kg, 7.5 mg/kg, and 15 mg/kg mitigated cyclophosphamide-induced leukopenia in a dose-dependent manner. The TLC reduction was least pronounced at 7.5 mg/kg, suggesting an optimal dose for mitigating leukopenia. Like lymphocyte count, monocyte count also showed a similar pattern, i.e., 7. 5 mg/kg dose having relatively higher counts post-treatment than the other doses. The same trend was evident with neutrophil counts as MXF treatment also influenced neutrophils dose-dependently. MXF 15 mg/kg treated group had the most significant effect. The exact

**Table 2. Survival rates (95% CI) of experimental groups in the mice lethality assay with different doses of MXF compared using Chi-square test. Level of significance was considered as p < 0.05.**

| Groups | Alive | Dead | Chi-Square |
|---|---|---|---|
| Negative Control | 5[100% (95%CI 56.55–100)] | 0[0% (95%CI 0–43.45)] | 0.0173 |
| Positive Control | 0[0% (95%CI 0–43.45)] | 5[100% (95%CI 56.55–100)] | |
| MXF 3.75 mg/kg | 3[60% (95%CI 23.07–92.89)] | 2[40% (95%CI 7.11–76.93)] | |
| MXF 7.5 mg/kg | 2[40% (95%CI 7.11–76.93)] | 3[60% (95%CI 23.07–92.89)] | |
| MXF 15 mg/kg | 1[20% (95%CI 1.03–62.45)] | 4[80% (95%CI 37.55–98.97)] | |

pathway through which MXF influences the immune response is not well understood; however, some possibilities have been postulated. It is believed that MXF may interfere with activating the nuclear factor kappa-light-chain-enhancer of activated B cells (NF-κB), a transcription factor that plays a crucial role in the inflammatory response [34]. Thus, MXF influences the downregulation of NF-κB and decreases the expression of pro-inflammatory cytokines, including TNF-α, IL-1β, and IL-6 [34,35]. Furthermore, MXF might interfere with the MAPK pathways associated with intracellular signal transduction in response to stress and inflammation [7,37]. MXF might suppress the activity of MAPK and thus lessen cytokine levels and inflammation [7]. These proposed mechanisms align with our findings since the treated mice revealed decreased pro-inflammatory cytokines and WBC count compared to the untreated group. MXF is a well-established antibiotic widely used against the treatment of the pneumonia [36] and acute exacerbation of the COPD [37]. MXF is reported to suppress the lymphocytic activity [38], however, the deduction that which cell types were affected by the MXF in this study is difficult. Furthermore, the in vitro assays performed in this study defines the anti-inflammatory effect of the drug by interfering with the cytokines as evident from previous studies and explained in the latter part of this study [39].

In our study, MXF significantly influences cytokine levels in a dose-dependent manner, enhancing anti-inflammatory cytokines while reducing pro-inflammatory cytokines. In vitro, TGF-β levels were highest at 32 µg/ml, IL-10 peaked at 16 µg/ml, IL-6 was highest at 4µg/ml, and TNF-α was significantly reduced at 64 µg/ml. These findings were corroborated in vivo, with the highest TGF-β levels observed in the MXF 15 mg/kg and positive control groups, while IL-10 levels peaked in the positive control group, followed by the MXF 3.75 mg/kg group. Additionally, IL-6 and TNF-α levels were significantly lower in the MXF 15 mg/kg group, indicating a potent anti-inflammatory effect at higher doses. The results concurred with other studies that contributed to the substantiation of MXF's anti-inflammatory properties. Wang, Wu (7) and Assar, Nosratabadi [[34] revealed that MXF could inhibit cytokine release by using isolated human peripheral blood mononuclear cells, which suppressed HUB/PMNs. Similarly, Sauer, Peukert [40] have reported that MXF reduced levels of TNF-α and IL-6 in a mouse model of acute lung injury. MXF has also been reported to reduce the TNF-α and IL-6 production in peripheral blood mononuclear cells stimulated with lipopolysaccharides (LPS) or heat-killed bacteria in an increasing concentration-dependent manner [41,42]. This decline in cytokine expression is linked to its action on CD14-positive cells that have the key role in the immune response to bacterial stimuli [43–45]. However, the immunomodulatory effect is not linked to cellular toxicity. Fluoroquinolones are believed to exert these effects by inhibiting mammalian topoisomerase type II enzyme particularly at supratherapeutic doses [46]. Other antibiotic classes, such as macrolides, lincosamides, and tetracyclines, have also been associated with modulating immune responses [44,47]. For example, ciprofloxacin has been found to enhance innate immunity by stimulating granulocyte-macrophage colony-stimulating factor (GM-CSF) production, which has been shown to improve pregnancy outcomes in certain immune conditions [48,49]. The reduction in the TNF-α and IL-1b by MXF exhibits its protective properties in sepsis and protects the organ damage. Furthermore, its antimicrobial activity is not associated with the DNA damage hence it may be further investigated for its protection in the sepsis, one of the leading causes of death in critically ill patients [50].

Our findings corroborated prior studies on FQs, in which similar immunomodulatory effects were observed. For example, ciprofloxacin increased neutrophil and macrophage function similarly to MXF [51–53]. Another study also revealed that levofloxacin increased IL-10 production and decreased TNF-α levels, similar to our findings [54]. However, some differences can be pointed out between the results of the present study and the previous research. A few research studies have described a more significant improvement in the immune status in response to fluoroquinolones at lower concentrations. For example, one study found that ciprofloxacin at higher doses significantly enhanced phagocytic activity, whereas our study observed a suppression of phagocytic activity at higher doses of MXF [55]. Additionally, while we found a significant reduction in TNF-α levels at higher doses of MXF, other studies reported no significant changes in TNF-α levels with other fluoroquinolones at comparable doses [56].

The changes observed in the counts of pro-inflammatory cytokines and WBC in the present study have essential clinical implications. Pathology is most often connected with an inflammatory response that raises the level of tissue injury and the time required for the repair process [57]. Therefore, by reducing such an inflammatory response, MXF could enhance clinical benefits in severely infected patients [58]. For instance, Liapikou and Torres [59] reported that the anti-inflammatory effect of MXF can be employed to mitigate lung irritation as ascribed to community-acquired pneumonia. The immunomodulatory effects are not peculiar to FQs but are expressed by other classes of antimicrobials. For example, azithromycin belongs to macrolides and has been established to have anti-inflammatory activity [60]. It reduces the synthesis of cytokines and prevents neutrophil chemotaxis [61]. The immunomodulating effects of macrolides and fluoroquinolones were studied by various researchers [35,62]. The findings demonstrated that both classes have anti-inflammatory effects. However, comparative analysis indicates that the immunomodulatory effect of MXF might be tremendously superior to that of non-fluoroquinolone antimicrobials [63].

The adverse events associated with the use of MXF are common but rarely serious. It is reported to cause the QT prolongation among children [64]. Studies have identified that the MXF enhances the sensitivity of the natural killer cells and result in their early activation and proliferation, hence may be used safely [7]. However, a few case studies have cautioned for the side effect of the MXF in patients with idiopathic thrombocytopenic purpura (ITP) [65]. Another case study also pointed out towards the thrombocytopenia induced by MXF, however, it was subsided after the discontinuation of the drug [66].Regarding the safety profile of MXF, the mice lethality assay revealed a significant dose-dependent increase in mortality among the groups (p = 0.0173), with 0% mortality in the negative control, 100% in the positive control, and 40%, 60%, and 80% in the groups treated with 3.75 mg/kg, 7.5 mg/kg, and 15 mg/kg of MXF, respectively. These results align with studies by Murray, Ikuta [67] and Miyake, Miyazaki [68], who observed similar dose-response relationships in their toxicity assessments. However, Nandanwar, Kansagara [69] reported no significant increase in mortality with higher doses.

## 5. Conclusions

This study demonstrates that MXF possesses significant immunomodulatory properties, effectively reducing cyclophosphamide-induced neutropenia and altering cytokine levels and phagocytic activity. The *in vitro* and *in vivo* assays exhibited that the MXF affect the synthesis of pro-inflammatory cytokines.Hence offering promising benefits for immunocompromised patients, caution is advised at higher doses due to potential safety concerns.

## 6. Limitations

The present study is one of the few that have investigated the immunomodulatory effects of MXF in a murine model by assessing key inflammatory cytokine levels. While the findings provide valuable insights into the immunological responses elicited by MXF, the study has certain limitations. Specifically, Toxicological and histopathological analyses were not the part of the study. The study also does not provide the precise molecular pathways and mechanisms through which MXF regulates the immune responses. Understanding the precise signaling mechanisms by MXF remains an area for further exploration. Additionally, a comprehensive safety evaluation was not conducted as part of this research. Future investigations should address these gaps to better understand the immunomodulatory actions and safety profile of MXF.

## Supporting information

**S1 Data. Statistical analyses data** .
(DOCX)

**S2 Data. Plos one humane endpoints checklist.**
(PDF)

## Author contributions

**Conceptualization:** Adeel Sattar, Mehmood Ahmad.

**Data curation:** Waqas Ahmad, Bilal Mahmood Beg.

**Formal analysis:** Rana Muhammad Zahid Mushtaq, Mostafa A. Abdel Maksoud.

**Investigation:** Mehmood Ahmad, Sammina Mahmood.

**Methodology:** Mehmood Ahmad, Abdur Rauf Khalid.

**Project administration:** Adeel Sattar, Mehmood Ahmad.

**Resources:** Adeel Sattar, Naeem Rasool, Abdur Rauf Khalid, Mostafa A. Abdel Maksoud, Abdul Aziz Alamri.

**Software:** Rana Muhammad Zahid Mushtaq, Waqas Ahmad.

**Supervision:** Adeel Sattar.

**Validation:** Waqas Ahmad, Bilal Mahmood Beg, Mostafa A. Abdel Maksoud.

**Visualization:** Naeem Rasool, Abdul Aziz Alamri.

**Writing – original draft:** Mehmood Ahmad, Waqas Ahmad.

**Writing – review & editing:** Sammina Mahmood, Rana Muhammad Zahid Mushtaq.

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
