## [Decision Letter · Decision Letter 0]

5 Nov 2024

PONE-D-24-39420Understanding the Impact of Moxifloxacin on Immune Function: Findings from Cytokine Analyses and Immunological Assays in MicePLOS ONE

Dear Dr. Sattar,

Thank you for submitting your manuscript to PLOS ONE. After careful consideration, we feel that it has merit but does not fully meet PLOS ONE’s publication criteria as it currently stands. Therefore, we invite you to submit a revised version of the manuscript that addresses the points raised during the review process.

Dear Author,

In this article, the authors explained the immunomodulatory effects of moxifloxacin on immune function by conducting invitro and in vivo analyses in Swiss mice. The experiments are well-designed, however, the manuscript needs a major revision.

Please look into the following comments:

1. The methodology section needs to be clearly rewritten with sub-sections explining clearly each sub-experiment carried out.

2. The objectives of the study should be presented clearly in the intoduction part coorelating with the experiments carried out.

3. Why only female mice were included in the study.

4. The authors mentioned in the discussion that "The mice lethality test results revealed a significant difference in mortality among the groups (p=0.0173). The group receiving 3.75 mg/kg of MXF had a 40% mortality rate, while the group treated with 7.5 mg/kg of MXF exhibited a 60% mortality rate. The highest dose of 15 mg/kg of MXF resulted in the highest mortality rate among the treatment groups, with 80% of the mice deceased" . How could the authors justify immunomodulatory effect of Moxifloxacin when mortality is reported at the tested doses.

5. The histopathological analysis of spleen would substantiate the results of the study. The authors need to justify why histopathological analysis was not carried out.

6. Apart from the above comments, the reviewers have given note worthy comments which the authors need to address for justifying the study.

We look forward to receiving your revised manuscript.

Kind regards,

GV Narasimha Kumar

Academic Editor

PLOS ONE

Journal Requirements: When submitting your revision, we need you to address these additional requirements. 1. Please ensure that your manuscript meets PLOS ONE's style requirements, including those for file naming. The PLOS ONE style templates can be found at https://journals.plos.org/plosone/s/file?id=wjVg/PLOSOne_formatting_sample_main_body.pdf and https://journals.plos.org/plosone/s/file?id=ba62/PLOSOne_formatting_sample_title_authors_affiliations.pdf 2. Please include your tables as part of your main manuscript and remove the individual files. Please note that supplementary tables (should remain/ be uploaded) as separate ""supporting information"" files 3. Please include captions for your Supporting Information files at the end of your manuscript, and update any in-text citations to match accordingly. Please see our Supporting Information guidelines for more information: http://journals.plos.org/plosone/s/supporting-information.

**Additional Editor Comments:**

Dear Author,

In this article, the authors explained the immunomodulatory effects of moxifloxacin on immune function by conducting invitro and in vivo analyses in Swiss mice. The experiments are well-designed, however, the manuscript needs a major revision.

Please look into the following comments:

1. The methodology section needs to be clearly rewritten with sub-sections explining clearly each sub-experiment carried out.

2. The objectives of the study should be presented clearly in the intoduction part coorelating with the experiments carried out.

3. Why only female mice were included in the study.

4. The authors mentioned in the discussion that "The mice lethality test results revealed a significant difference in mortality among the groups (p=0.0173). The group receiving 3.75 mg/kg of MXF had a 40% mortality rate, while the group treated with 7.5 mg/kg of MXF exhibited a 60% mortality rate. The highest dose of 15 mg/kg of MXF resulted in the highest mortality rate among the treatment groups, with 80% of the mice deceased" . How could the authors justify immunomodulatory effect of Moxifloxacin when mortality is reported at the tested doses.

5. The histopathological analysis of spleen would substantiate the results of the study. The authors need to justify why histopathological analysis was not carried out.

6. Apart from the above comments, the reviewers have given note worthy comments which the authors need to address for justifying the study.

Regards,

Dr Narasimha Kumar GV

Reviewers' comments:

Reviewer's Responses to Questions

**Comments to the Author**

1. Is the manuscript technically sound, and do the data support the conclusions?

Reviewer #1: Partly

Reviewer #2: Yes

2. Has the statistical analysis been performed appropriately and rigorously? 

Reviewer #1: Yes

Reviewer #2: Yes

3. Have the authors made all data underlying the findings in their manuscript fully available?

Reviewer #1: Yes

Reviewer #2: No

4. Is the manuscript presented in an intelligible fashion and written in standard English?

Reviewer #1: Yes

Reviewer #2: Yes

5. Review Comments to the Author

Reviewer #1: 1. It is advised that the effect of MXF’s immunomodulatory properties should be explained, especially in inflammatory and immune-related disorders.

2. Give the relevance why only female mice were used in the study.

3. On what bases, the dose of MXF (3.75 mg/kg, 7.5 mg/kg, 15 mg/kg) was selected for the study.

4. Give the concentration of SRBCs used for the study with proper justification and reference.

5. Give the details about positive control group. There is no clear details what intervention was given to positive control group.

6. There is lack of reference group in the study. The methodology is not clear whether author used standard drug or not.

7. Authors should give more detailed theories about how MXF might change the amounts of cytokines like TNF-alpha and IL-6 would be helpful for the study.

8. Even though the primary focus of the study is MXF, however, a quick comparison with other FQs could help emphasize its special qualities and determine whether the effects are unique to MXF or a more general trait of FQs.

9. It is well known that antibiotics' immunomodulatory actions can occasionally have negative side effects, which can exacerbate immunosuppression or autoimmune. Author should provide the safety parameters of the study.

10. The authors should provide the relevance and applicability of the research in the context of the immunomodulatory actions of MXF in human beings.

11. The statistical part in the legends of each figure is not mentioned lacking the clarity.

Reviewer #2: In this article, the authors explain the impact of moxifloxacin on immune function by conducting invitro and in vivo analyses in mice. The experiment is well-designed, results are presented appropriately. However, the manuscript needs a major revision. Please see specific comments below.

Introduction

Line 44-Expand MXF when used first in the body of the article.

Line 56-MXF was also 56 found to suppress the host immunity while enhancing it [9]. Please re write the sentence. The meaning is not clear.

Materials and methods

Provide a detailed separate animal experiment protocol used in the study at the beginning of this section. How many animals were used in the study? Age/ strain? Please include details on management of experimental animals. Illustrate design of animal experimentation in a table or a diagram. How many animals were included in the control group? How many treatment groups were there? What treatments were given in each group? Please refer some similar experiments in mouse models like-A cyclophosphamide-induced immunosuppression Swiss Albino mouse model unveils a potential role for cow urine distillate as a feed additive. Journal of Ayurveda and Integrative Medicine.

Each experiments (in vitro and in vivo) were performed to meet different objectives of the study This need to be clearly mentioned in the methods section.

To be specific, this is given in the abstract. (2) Methods: Swiss female albino 25 mice were treated with different concentrations of MXF, and the immunological studies were performed using a cytokine assay, carbon clearance test, indirect hemagglutination test, and a mice lethality assay. Please mention how each methods were done in your experiment clearly in different paragraphs if possible. After the initial animal experimentation design section, Begin next section with clear lead sentence. For example, for performing cytokine assay (explain what you have done). Then, carbon clearance test (explain what you have done).

Please include some references in the methods section to explain the animal experimentation and experimental design.

Stat analysis

Please describe the nature of data obtained in each experiment. Were any data transformation performed?

Line 131-Please add the level of significance set for ANOVA (p < 0.05). Are the raw data made available? If so, please mention.

Results

The results section follows a sequential pattern with easy to comprehend illustration. Follow the same in methods section also.

The figures and tables are given at the end of the para of each results section. Please put it in the beginning sentences. It is difficult to comprehend the lengthy results in each para of the text. Use the figures to explain the results in the text. For example, as illustrated in the figure 1 (explain your results).

Discussion

Good narration. Please consider the following.

Line 230-

The exact pathway through which MXF influences the immune response is not well understood; however, some possibilities have been postulated- Explain which are those possibilities? Please add appropriate references.

Reference 28-Could you please add research articles also to substantiate this?

Conclusion

I would add a concluding sentence like, by doing invitro and invivo analyses, the study found…..

References

Please have a thorough language check. Please format the references in line with standard formatting. For example, in reference, italicise scientific names. For instance, Mycobacterium tuberculosis in reference 5 and 11.

Reference 8-Please give the complete reference.

Reference 9-Please add additional details here to connect the significance of your study illustrating how your study fills the gap in our understanding. For example, I would include, this previous study got those results.

6. PLOS authors have the option to publish the peer review history of their article (what does this mean? ). If published, this will include your full peer review and any attached files.

**Do you want your identity to be public for this peer review?** For information about this choice, including consent withdrawal, please see our Privacy Policy .

Reviewer #1: No

Reviewer #2: No

---

## [Author Response · Author response to Decision Letter 0]

16 Jan 2025

Responses to the Reviewers

Thank you for your careful and professional and productive suggestions that greatly help us to improve our manuscripts. We have carefully revised the manuscript according to your comments. The responses to the comments are:

Comments and Suggestions for Authors

In this article, the authors explained the immunomodulatory effects of moxifloxacin on immune function by conducting invitro and in vivo analyses in Swiss mice. The experiments are well-designed; however, the manuscript needs a major revision.

Please look into the following comments:

Comment 1: The methodology section needs to be clearly rewritten with sub-sections explaining clearly each sub-experiment carried out.

Response: Thank you for your valuable feedback. The authors have added the sub-sections under the material and method section and has re-structured the section entirely to better align with the study.

Comment 2: The objectives of the study should be presented clearly in the introduction part correlating with the experiments carried out.

Response: We are thankful to the reviewers for their feedback. We have added the objectives of each assay that was mentioned in the materials and methods section. Each assay is now highlighted with their headings. Line 110. Line 120-121, Line 125, Line 138-139, Line 141-144

Comment 3. Why only female mice were included in the study.

Response: Thank you for query. Female mice were chosen to reduce variability and potential error in the experimental outcomes. Male mice often exhibit greater variability in behavior and physiological responses due to factors like aggression and stress, which could confound the results. Similar approach was adopted by the previous study (DOI: https://doi.org/10.1211/jpp.60.1.0008)

Comment 4. The authors mentioned in the discussion that "The mice lethality test results revealed a significant difference in mortality among the groups (p=0.0173). The group receiving 3.75 mg/kg of MXF had a 40% mortality rate, while the group treated with 7.5 mg/kg of MXF exhibited a 60% mortality rate. The highest dose of 15 mg/kg of MXF resulted in the highest mortality rate among the treatment groups, with 80% of the mice deceased". How could the authors justify immunomodulatory effect of Moxifloxacin when mortality is reported at the tested doses.

Response: Thank you for your valuable feedback. While we acknowledge the observed mortality at the 3.75 mg/kg dose. Our results, supported by statistical analyses, indicates that mortality at this dose was significantly lower compared to the positive control group and the higher MXF doses. This suggests a potential dose-dependent immunomodulatory effect of Moxifloxacin, with the lower dose mitigating the severity of infection-induced mortality to a greater extent.

Comment 5. The histopathological analysis of spleen would substantiate the results of the study. The authors need to justify why histopathological analysis was not carried out.

Response: Since this study was an early phase study and already encompasses various immunomodulatory experiments such as ELISA and other in vivo models so histopathology was not carried out in this study due to some technical limitations. The absence on histopathological study is the limitation of this study and may be taken up in the future aspects.

6. Apart from the above comments, the reviewers have given note worthy comments which the authors need to address for justifying the study.

Reviewer 1

Comment 1. It is advised that the effect of MXF’s immunomodulatory properties should be explained, especially in inflammatory and immune-related disorders.

Response: We appreciate this suggestion. We have expanded the discussion to include detailed explanations of MXF’s immunomodulatory properties, particularly in the context of inflammatory and immune-related disorders. We have referred to studies exploring the effects of MXF on cytokine production and immune cell modulation. Although there is little evidence of the use of MXF in the immune related disorders but it provides a dual response in the treatment of COPD or community acquired pneumonia (Line 277-310).

Comment 2. Give the relevance why only female mice were used in the study.

Response: Thank you for query. Female mice were chosen to reduce variability and potential error in the experimental outcomes. Male mice often exhibit greater variability in behavior and physiological responses due to factors like aggression and stress, which could confound the results. Similar approach was adopted by the previous study (DOI: https://doi.org/10.1211/jpp.60.1.0008)

Comment 3: On what bases, the dose of MXF (3.75 mg/kg, 7.5 mg/kg, 15 mg/kg) was selected for the study.

Response: The literature was reviewed for the doses of MXF. A standard dose of 400 mg is used up in the adult so the dose in the mice was extrapolated at 7.5 mg/kg as per the literature. Furthermore, we carried out a pilot study before the start of the experiment to define the doses at 3.75 mg/kg and 15 mg/kg in addition to 7.5 mg/kg.

Comment 4: Give the concentration of SRBCs used for the study with proper justification and reference.

Response: A higher concentration (1%) of SRBCs would increase the sensitivity of the assay by providing more targets for agglutination. Some previous studies have used the similar concentrations of SRBCs.

[Kolathingal-Thodika N, Usha PT, Sujarani S, Suresh NN, Priya PM, Naseef PP, Kuruniyan MS, Ollakkode S, Elayadeth-Meethal M. A cyclophosphamide-induced immunosuppression Swiss Albino mouse model unveils a potential role for cow urine distillate as a feed additive. Journal of Ayurveda and Integrative Medicine. 2023 Sep 1;14(5):100784.]

[Hamdani DA, Javeed A, Ashraf M, Nazir J. Evaluation of ketoprofen effects on humoral immunity and immune organs in mice. Pakistan Journal of Zoology. 2014 Dec 1;46(6).]

Comment 5. Give the details about positive control group. There is no clear details what intervention was given to positive control group.

Response: Positive group only received the neutropenic dose of cyclophosphamide but did not receive the MXF treatment. We have updated the details under methodology section at line # 108 to 110.

Comment 6: There is lack of reference group in the study. The methodology is not clear whether author used standard drug or not.

Response: The reference or control group was kept in every assay and the control group only received the solvent and no MXF was given to the control group.

Comment 7. Authors should give more detailed theories about how MXF might change the amounts of cytokines like TNF-alpha and IL-6 would be helpful for the study.

Response: Thank you for your valuable feedback. In response, we have incorporated additional information from the literature to provide a more detailed explanation of how moxifloxacin (MXF) influences cytokine production, specifically TNF-α and IL-6. The updated section (lines 257–266) elaborates on MXF's immunomodulatory effects, including its concentration-dependent inhibition of cytokine production and the potential mechanisms underlying these effects

Comment 8. Even though the primary focus of the study is MXF, however, a quick comparison with other FQs could help emphasize its special qualities and determine whether the effects are unique to MXF or a more general trait of FQs.

Response: The authors have included this comparison of the FQs at various places such as Line 276-288.

Comment 9. It is well known that antibiotics' immunomodulatory actions can occasionally have negative side effects, which can exacerbate immunosuppression or autoimmune. Author should provide the safety parameters of the study.

Response: Thank you for highlighting it. We acknowledge that the immunomodulatory actions of antibiotics, including their potential to exacerbate immunosuppression or autoimmune conditions, are well-documented concerns. We also acknowledged the importance of addressing the potential negative side effects of MXF’s immunomodulatory actions. We also referenced similar studies that have evaluated the safety of antibiotics with immunomodulatory properties. We have included a section discussing the safety parameters (Line 333-351). However, our study primarily focused on evaluating the immunological and cytokine responses induced by Moxifloxacin in mice, we did not conduct a comprehensive safety evaluation as part of the experimental design. We have added this as a limitation of our study.

Comment 10. The authors should provide the relevance and applicability of the research in the context of the immunomodulatory actions of MXF in human beings.

Response: Thankful for this insightful comment. We have added a suggestion enhancing the discussion, however, since there is very less data highlighting the immunomodulatory action of MXF in human beings as of now (Line 310). We have added it in revised manuscript under the conclusion section as well. Our study provides foundational insights into the immunomodulatory effects MXF in a controlled murine model. “The in vitro and in vivo assays exhibited that the MXF affect the synthesis of pro-inflammatory cytokines offering promising benefits for immunocompromised patients, however, caution is advised at higher doses due to potential safety concerns. “Although, direct extrapolation to humans should be approached with caution due to interspecies differences.

Comment 11. The statistical part in the legends of each figure is not mentioned lacking the clarity.

Response: Authors are thankful for highlighting this short coming. We have updated the legends in figures.

Reviewer 2

Reviewer #2: In this article, the authors explain the impact of moxifloxacin on immune function by conducting invitro and in vivo analyses in mice. The experiment is well-designed, results are presented appropriately. However, the manuscript needs a major revision. Please see specific comments below.

Introduction

Comment 1: Line 44- Expand MXF when used first in the body of the article.

Response: We are thankful to the reviewer for pointing this out. We have added the text prior to MXF.

Comment 2: Line 56-MXF was also found to suppress the host immunity while enhancing it [9]. Please re write the sentence. The meaning is not clear.

Response: Thank you for the comment. We have expanded the MXF at the first appearance in the manuscript. Additionally, we have rewritten the sentence expanding form line 56 to 59.

Comment 3: Materials and methods Provide a detailed separate animal experiment protocol used in the study at the beginning of this section. How many animals were used in the study? Age/ strain? Please include details on management of experimental animals. Illustrate design of animal experimentation in a table or a diagram. How many animals were included in the control group? How many treatment groups were there? What treatments were given in each group? Please refer some similar experiments in mouse models like-A cyclophosphamide-induced immunosuppression Swiss Albino mouse model unveils a potential role for cow urine distillate as a feed additive. Journal of Ayurveda and Integrative Medicine.

Response: Thank you for highlighting the shortcoming. A total of 95 mice were used in the study. We have added a graphic illustration of the workflow. Additionally, the suggested reference was much helpful in this regard. We have cited it in the reference list.

Comment 4: Each experiments (in vitro and in vivo) were performed to meet different objectives of the study This need to be clearly mentioned in the methods section.

To be specific, this is given in the abstract. (2) Methods: Swiss female albino 25 mice were treated with different concentrations of MXF, and the immunological studies were performed using a cytokine assay, carbon clearance test, indirect hemagglutination test, and a mice lethality assay. Please mention how each methods were done in your experiment clearly in different paragraphs if possible. After the initial animal experimentation design section, Begin next section with clear lead sentence. For example, for performing cytokine assay (explain what you have done). Then, carbon clearance test (explain what you have done).

Response: We appreciate this feedback and have restructured the methods section accordingly. Each experiment is now described in separate paragraphs, with clear lead sentences outlining the specific objectives and detailed procedures. We hope this gives clarity to our methodology.

Comment 5: Please include some references in the methods section to explain the animal experimentation and experimental design.

Response: We have added relevant references to support our experimental design and procedures. (Line 93-102)

Stat analysis

Comment 6: Please describe the nature of data obtained in each experiment. Were any data transformation performed? Line 131-Please add the level of significance set for ANOVA (p < 0.05). Are the raw data made available? If so, please mention.

Response: The data was under normality conditions as checked by Kolmogorov-Smirnov test. Therefore, transformation was not considered except IHA titers which were subjected to Log2 transformation. At line No. 131 we have now updated the level of significance set for ANOVA (p < 0.05). The raw data has been made available.

Results

Comment 7: The results section follows a sequential pattern with easy to comprehend illustration. Follow the same in methods section also.The figures and tables are given at the end of the para of each results section. Please put it in the beginning sentences. It is difficult to comprehend the lengthy results in each para of the text. Use the figures to explain the results in the text. For example, as illustrated in the figure 1 (explain your results).

Response: Thank you for your constructive suggestion regarding the placement of figures and tables within the results section. We agree that integrating figures at the beginning of each paragraph and using them to explain the results can significantly enhance the clarity and readability of the text. We have revised the manuscript accordingly by reordering the figures and tables and restructuring the results section

Discussion

Good narration. Please consider the following.

Line 230-

Comment 8: The exact pathway through which MXF influences the immune response is not well understood; however, some possibilities have been postulated- Explain which are those possibilities? Please add appropriate references.

Response: The possible pathways through which MXF influences the immune response are enlisted from (Line 269-275). The supported references to studies are:

- Wang M, Wu H, Jiang W, Ren Y, Yuan X, Wang Y, et al. Differences in nature killer cell response and interference with mitochondrial DNA induced apoptosis in moxifloxacin environment. International Immunopharmacology. 2024;132:111970.

- Assar S, Nosratabadi R, Khorramdel Azad H, Masoumi J, Mohamadi M, Hassanshahi G. A review of immunomodulatory effects of fluoroquinolones. Immunological Investigations. 2021;50(8):1007-26

- Dawood JO, Abu-Raghif A. Moxifloxacin's Therapeutic Effects in AA-Induced Colitis: Anti-Inflammatory Action through NF-κB Pathway Inhibition, Including TNF-α Pathway and Downstream Inflammatory Processes. Journal of Contemporary Medical Sciences. 2023;9(4)

Comment 9: Reference 28-Could you please add research articles also to substantiate this?

Response: We have added the additional research articles to substantiate the claims made in reference 28:

1. Cheng, X., Wang, C., Su, Y., Luo, X., Liu, X., Song, Y. and Deng, Y., 2018. Enhanced opsonization-independent phagocytosis and high response ability to opsonized antigen–antibody complexes: a new role of kupffer cells in the accelerated blood clearance phenomenon upon repeated injection of PEGylated emulsions. Molecular Pharmaceutics, 15(9), pp.3755-3766.

2. Moektiwardoyo M, Rositah S, Kusuma AF. Immunomodulatory activity of Plectranthus scutellarioides (L.) R. Br. leaves ethanolic extract and its fraction on rat using carbon clearance method. Drug Invention Today. 2019 Nov 1;11(11).

3. Dillasamola DW, Aldi YU, Fakhri MU, Diliarosta S, Biomechy OP. Immunomodulatory effect test from moringa leaf extract (Moringa oleifera L.) with carbon clearance method in male white mice. Asian J Pharmac

---

## [Decision Letter · Decision Letter 1]

2 Mar 2025

PONE-D-24-39420R1Understanding the Impact of Moxifloxacin on Immune Function: Findings from Cytokine Analyses and Immunological Assays in MicePLOS ONE

Dear Dr. Sattar,

Thank you for submitting your manuscript to PLOS ONE. After careful consideration, we feel that it has merit but does not fully meet PLOS ONE’s publication criteria as it currently stands. Therefore, we invite you to submit a revised version of the manuscript that addresses the points raised during the review process. 

We look forward to receiving your revised manuscript.

Kind regards,

GV Narasimha Kumar

Academic Editor

PLOS ONE

Journal Requirements:

Additional Editor Comments:

Dear Author,

The revised manuscript has addressed major queries of both the editor and reviewers. However, a few minor revisions need to be done to make the manuscript suitable for publication. The minor suggestions are as follows:

1. Add the date of approval of the Animal Ethics Committee

2. Have a proof-reading for correcting minor language edits

3. Avoid repeated use of Moxifloxacin, as the abbreviation MXF is enough. For example, lines 48 and 51 in the introduction section. Please re-write the sentences to avoid starting the sentences with MXF. Please avoid using capital letters in between sentences. For example, lines 333 and 343 in the limitation section. Please be consistent in the usage of the abbreviation MXF and full-form moxifloxacin in the manuscript. I would use the full form for the first time in the abstract and body of the text, then only MXF throughout.

4. In line 322 of the last paragraph of the discussion, add a space after a period.

Figures

1. Edit the figures and make them uniform. For example, please use the same font for all labels within the figures. In Figure 1, Control and control are both used. Please be consistent. Re-write the legends for figures to make them grammatically correct.

Regards,

Dr Narasimha Kumar

Reviewers' comments:

Reviewer's Responses to Questions

**Comments to the Author**

1. If the authors have adequately addressed your comments raised in a previous round of review and you feel that this manuscript is now acceptable for publication, you may indicate that here to bypass the “Comments to the Author” section, enter your conflict of interest statement in the “Confidential to Editor” section, and submit your "Accept" recommendation.

Reviewer #2: All comments have been addressed

Reviewer #3: All comments have been addressed

2. Is the manuscript technically sound, and do the data support the conclusions?

Reviewer #2: Yes

Reviewer #3: Yes

3. Has the statistical analysis been performed appropriately and rigorously? 

Reviewer #2: Yes

Reviewer #3: Yes

4. Have the authors made all data underlying the findings in their manuscript fully available?

Reviewer #2: Yes

Reviewer #3: Yes

5. Is the manuscript presented in an intelligible fashion and written in standard English?

Reviewer #2: Yes

Reviewer #3: Yes

6. Review Comments to the Author

Reviewer #2: I have added a few more minor corrections.

Please add the date of approval of the Animal Ethics Committee

Please have a proof-reading for correcting minor language edits

Please avoid repeated use of Moxifloxacin, as the abbreviation MXF is enough. For example, lines 48 and 51 in the introduction section. Please re-write the sentences to avoid starting the sentences with MXF. Please avoid using capital letters in between sentences. For example, lines 333 and 343 in the limitation section. Please be consistent in the usage of the abbreviation MXF and full-form moxifloxacin in the manuscript. I would use the full form for the first time in the abstract and body of the text, then only MXF throughout.

In line 322 of the last paragraph of the discussion, add a space after a period.

Figures

Please edit the figures and make them uniform. For example, please use the same font for all labels within the figures.

In Figure 1, Control and control are both used. Please be consistent. I would prefer Control.

Please re-write the legends for figures to make them grammatically correct.

Reviewer #3: (No Response)

7. PLOS authors have the option to publish the peer review history of their article (what does this mean? ). If published, this will include your full peer review and any attached files.

**Do you want your identity to be public for this peer review?** For information about this choice, including consent withdrawal, please see our Privacy Policy .

Reviewer #2: **Yes: ** Muhammed Elaaydeth Meethal

Reviewer #3: **Yes: ** Dr. Vara Prasad Saka

---

## [Author Response · Author response to Decision Letter 1]

12 Mar 2025

Dear Editor,

We appreciate the constructive feedback provided by you and the reviewers. We have carefully addressed all the minor revisions as suggested. Below is our point-by-point response to each comment:

Comment 1: Review the reference list to ensure it is complete and correct. Remove retracted papers or provide a rationale for their inclusion. If cited, mark retracted references appropriately.

Response: We have thoroughly reviewed our reference list. No retracted references were included in our manuscript.

Comment 2: Add the date of approval of the Animal Ethics Committee.

Response: The date of approval from the Animal Ethics Committee has been added in the manuscript.

Comment 3: Proofread the manuscript for minor language corrections.

Response: We have carefully proofread the manuscript and corrected minor language issues.

Comment 4: Avoid repeated use of "Moxifloxacin," as the abbreviation "MXF" is sufficient. Use the full form only once in the abstract and body, then "MXF" throughout. Avoid starting sentences with abbreviations and capitalize words appropriately.

Response: We have revised the manuscript to ensure that "Moxifloxacin" appears in full form only the first time in the abstract and body followed by "MXF" throughout.

Sentences have been rewritten to avoid starting with abbreviations and capitalization inconsistencies have been corrected.

Comment 5: In line 322 of the last paragraph of the discussion, add a space after a period.

Response: The formatting issue has been corrected by adding the necessary space.

Comment 6: Ensure consistency in figures, including uniform fonts and labels. In Figure 1, standardize the use of “Control” vs. “control.”

Response: The figures have been revised. In Figure 1, "Control" has been used consistently.

Comment 7: Re-write the figure legends for grammatical correctness.

Response: The figure legends have been revised for grammatical accuracy and clarity.

We are thankful for the opportunity to improve our manuscript and thank you for your time and consideration.

---

## [Editor Report · Decision Letter 2]

14 Mar 2025

Understanding the Impact of Moxifloxacin on Immune Function: Findings from Cytokine Analyses and Immunological Assays in Mice

PONE-D-24-39420R2

Dear Dr. Sattar,

We’re pleased to inform you that your manuscript has been judged scientifically suitable for publication and will be formally accepted for publication once it meets all outstanding technical requirements.

Kind regards,

GV Narasimha Kumar

Academic Editor

PLOS ONE

Additional Editor Comments (optional):

Dear Author,

I am happy to let you know that the manuscript is now technically suitable for publication in the present form.

Regards,

Dr Narasimha Kumar
---

## [Editor Report · Acceptance letter]

PONE-D-24-39420R2

PLOS ONE

Dear Dr. Sattar,

I'm pleased to inform you that your manuscript has been deemed suitable for publication in PLOS ONE. Congratulations! Your manuscript is now being handed over to our production team.

Kind regards,

on behalf of

Dr. GV Narasimha Kumar

Academic Editor

PLOS ONE